# Black Locust (*Robinia pseudoacacia* L.) in Romanian Forestry

Alexandru Liviu Ciuvăț [1] , Ioan Vasile Abrudan [2] , Cristiana Georgeta Ciuvăț [1], Cristiana Marcu [1], Adrian Lorenăț [1], Lucian Dincă [1] and Bartha Szilard [3,*]

[1] National Institute for Research and Development in Forestry "Marin Drăcea", 077190 Bucharest, Romania
[2] Department of Silviculture, Faculty of Silviculture and Forest Engineering, Transilvania University of Brasov, 500123 Brașov, Romania
[3] Department of Forestry and Forest Engineering, University of Oradea, 410048 Oradea, Romania
* Correspondence: barthaszilard10@yahoo.com; Tel.: +40-727307277

**Abstract:** This paper presents a literature review of black locust (*Robinia pseudoacacia* L.) and the knowledge accumulated by Romanian foresters and researchers, covering species propagation, stand management, and vulnerability issues. As highlighted by numerous authors, black locust manifests dual features, both as an exogenous species and one that is already naturalized. The main drivers for this species' expansion in Romania is its ecological adaptability on degraded lands, fast growth, and high biomass yields, in addition to other economic benefits. Black locust plantations and coppices also offer an important range of ecosystem services such as $CO_2$ sequestration, landscape reclamation, fuel wood, or maintaining traditional crafts in regions with little to no forest cover. Highlighted disadvantages include short lifespan, invasiveness when introduced on fertile sites, and dieback in drought/frost prone areas. The results of extensive research and studies are captured in technical norms, although aspects such as species genetics, invasive potential, and adaptation to climate change dynamics call for more research and optimizing in species management. As Romania rallies its efforts with those of the international community in order to address climate change and desertification, black locust stands out as a proven solution for reclaiming degraded lands when native species are not an alternative.

**Keywords:** black locust; silviculture; ecology; management; risks; uses; impact

## 1. Introduction

Black locust (*Robinia pseudoacacia* L.) is a North American species that was introduced in Romania around the end of the 17th century [1], and it was first used in large-scale afforestation of degraded lands in the year 1852 [2]. Due to its remarkable adaptability, fast growth, and vigorous sprouting capacity, it has become one of the most widely spread exotic species in Romania [3]. Using black locust for afforestation, Romanian foresters successfully reclaimed large areas of abandoned agricultural land degraded by "flying sands" in southwest of the country [4], thus contributing to mitigation of the aridization phenomena [4].

Driving the extensive use of the species was its durable and versatile wood, much appreciated by rural communities, complemented with major benefits from crop fields and settlement protection against wind/sand deflation [5], as well as economically viable byproducts. In addition to the immediate improvements to local microclimate, afforestations also contributed to long-term climate change mitigation by sequestering atmospheric $CO_2$ [6–9] in the carbon pools (living tree biomass, soil organic matter, litter) as well as downstream wood products (e.g., furniture). Nevertheless, in the past decade, awareness has been raised towards its invasive potential in protected areas.

The authors aim at highlighting the peculiarities of this species in Romanian forestry with regard to its ecology and management, but also its economic importance and potential environmental impact.

## 2. Materials and Methods

This work aims for a comprehensive review of Romanian research and studies about the silviculture, management, and impact of black locust. The literature on the subject was found by consulting the archive of the National Institute for research and Development in Forestry "Marin Dracea" (approx. 150 publications), and by using the Google search engine. The largest pool of publications occurred at a national level between 1950 and 1990, regarding the silviculture and genetics of the species. The literature survey shows a poor post-1990 focus on this species, with a limited number of papers, especially regarding management, damaging factors, and risks.

## 3. Results

### 3.1. Black Locust in Romania—Areal and Ecological Behavior

*Robinia pseudoacacia* is a fast-growing species, reaching heights above 30 m and ages in excess of 100 years, while maturing at early ages of 5–7 years old. It regenerates both from stump sprouts and root suckers, and it can be easily propagated by cutting or grafting, although in Romanian forestry natural regeneration from root suckers and plantation of seedlings are almost exclusively used. It is reported to become invasive if inadequately planted in sites with high productivity potential [10], but also on poor sites due to its intolerance to other large tree species, so much that it forms pure stands. Nevertheless, it was also reported in association with shrubs and a few other compatible tree species whenever adequate planting schemes are applied [10,11]. Being a sun loving species, it prefers areas with long summers (mean annual temperatures above 10 °C), and annual rainfall between 400 and 600 mm. According to Ivanschi et al. [12] it grows well on sandy and sandy loam soils, with loose to lightly compaction, deep, with a medium humidity regime; conversely, it does not grow/survive on compact soils with calcium carbonates or soluble salts in top layers of soil or in the case of excess of humidity. Calcium carbonates ($CaCO_3$) in the top layers of soil (<40 cm) inhibit growth. Due to its nutritional particularities, it fixes nitrogen (N) in the soil (in symbiosis with N fixing bacteria) but it also consumes large quantities of soil minerals. It has a hard and durable wood, comparable with that of oak, which makes it very appreciated by rural populations.

Geographically, black locust stands occupy about 5% of the national forested area (250,000 ha), concentrated in the southwest, west, and in the east of the country [13].

The largest areas of compact stands and also the most valuable ones are located in the southwest of the country in the Oltenia region (Figure 1), which was also the location of its first introduction in Romania. Currently, the species is considered as a sub-spontaneous one, its range spreading from the plains to the lower mountain regions [3].

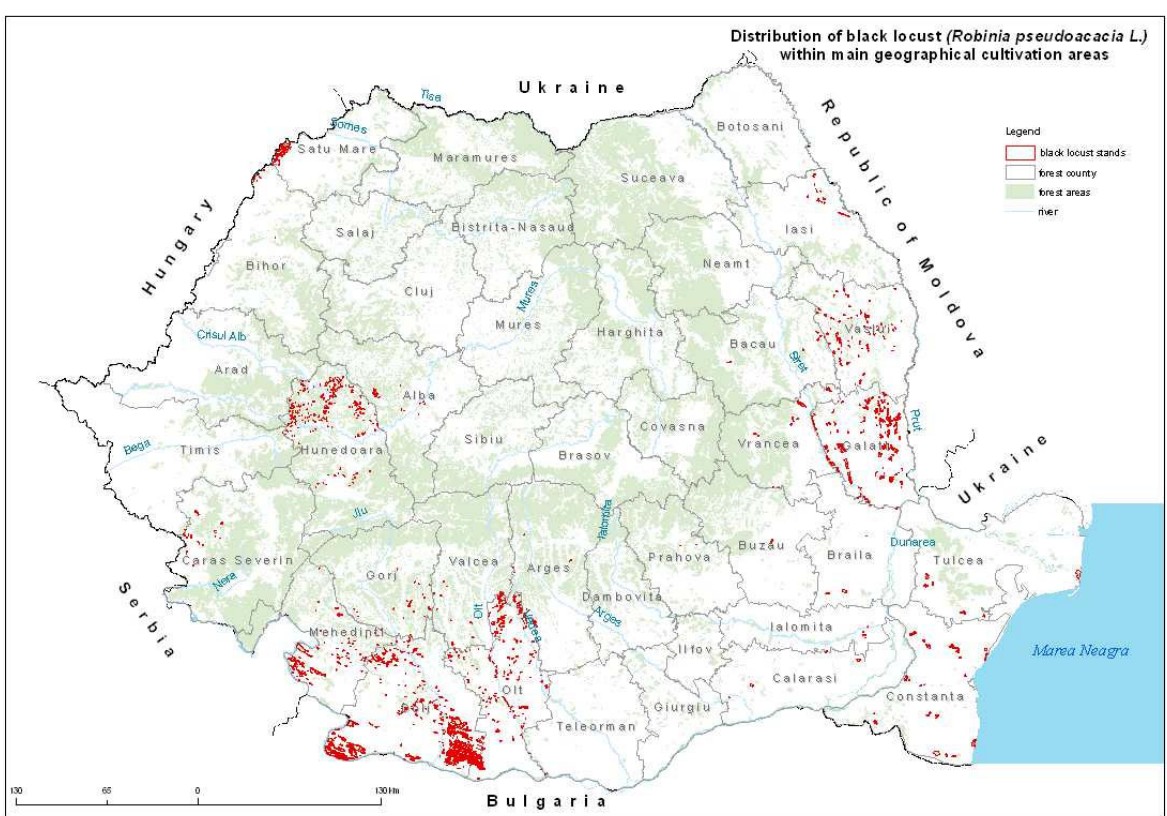

**Figure 1.** Distribution of black locust stands in the main culture regions of Romania [10].

*3.2. Overview of Management Approaches*

3.2.1. Forest Regeneration and Establishment of Plantations

Although black locust produces annually a very large number of seeds, because of their thick tegument, successful natural stand regeneration is missing almost completely [13]. Exceptionally isolated trees can be recognized as being from seeds in very particular sites, such as upper sides of river banks [14] or at pile burning sites where forest residues are burned [15]. Therefore, vegetative regeneration of harvested stands by mechanical stimulation of root sprouting and artificial regeneration by plantation of nursery seedlings in reforestation and afforestation of non-forest lands are preferred and largely practiced across the country [2].

3.2.2. Vegetative Regeneration

While the harvesting technique is always clear cutting, stand regeneration occurs via sprouts on the aboveground part of the stump or suckers by mechanical stimulation of roots by: (a) ploughing of the soil among the stumps at 20 cm depth; or (b) digging out the stump together with the roots around it in a radius of 50 cm combined with additional ploughing. When regeneration targets root-suckers, the sprouts must be removed through early tending operations, so as to not overwhelm the slow growing root suckers [13]. After two cycles of vegetative regeneration from sprouts, the stumps weaken and lose their sprouting capacity (due to fungal impact and soil nutrient depletion); and by the third generation, the stand's productivity drops to 40% of that of plantations [16]. That is why, after a second generation of sprouts, regeneration must be ensured from root-suckers [17]. In any case, after the third generation of vegetative regeneration, all of the stumps must be removed from the harvested area and further regeneration is ensured via the planting of seedlings. Recent studies have showed that, after clearcutting and stump removal, the density of root suckers can reach 50,000 plants per hectare [13].

### 3.2.3. Generative Regeneration

The recommended technique for seedling production includes harvesting the seeds directly from trees (late in the winter or early January) or separated from litter and mechanical or chemical 'forcing' before sowing next spring. Seedlings are almost without exception ready for transplantation after one season of growth (e.g., root collar diameter of 4 mm). After a second year in the nursery, they generally surpass the planting dimension standards which are associated with substantial risk of not surviving summer drought. Although, for afforestation in such areas, specific standards apply to ensure more robust plants are used—e.g., root collar diameter of 8 mm [17]. The seeds for nursery production of seedlings are often collected from seed orchards and have to meet standard requirements for purity, technical germination, and mass [18].

### 3.2.4. Afforestation Techniques

According to technical norms in force in Romania [19], afforestation with black locust is exclusively carried out by planting bare-root seedlings in hand-dug holes. This method has proved so successful in practice that there has been no more research on other methods, although associated costs may be prohibitive in an expensive labor market. The literature states that sowing the previously treated seeds can be used to create black locust stands [13]. Planting can be done both in spring as in autumn. Spring planting ensures adequate soil humidity (common in upper altitude regions), while planting in autumn (usually in the southern ad low-altitude regions) ensures root growth over winter and avoids spring flooding and early droughts [20].

The technical norms state that, in the case of sandy soils or degraded lands, optimum planting density for black locust is 5000 seedlings per hectare—with 2.0 m of distance between rows and 1.0 m between seedlings—and the planting pit is 40 cm wide and 40 cm deep. In the case of non-degraded lands, the planting density for black locust is 4000 seedlings per hectare—with 2.0 m of distance between rows and 1.25 m between seedlings—and the planting pit width, length, and depth is 40 cm. In order to ensure the highest survival rate, a key technical intervention after planting seedlings is trimming of their stems 1 cm above the soil before growing season starts [20]. Further on, to ensure success of plantations, regular maintenance involves gap filling in the first 2–3 years after planting, if necessary. The survival rate of plantations is checked yearly, with dead seedlings being replaced until canopy closure (2–4 years).

### 3.2.5. Tending Operations

Being a fast-growing species, canopy closure takes place early (1–3 years following planting), therefore tending operations (especially cleaning and thinning) need to be made at shorter periods of time (3–5 years) depending on the tree origin (seedlings or root-suckers) and density of the stand. Periodicity of tendings varies in function of stand development (size of trees) and influences its productivity (stand density influences the rate of growth) as well as its protective function, in which case higher stand densities are preferable only on degraded lands due to the slenderness coefficient [17].

Research and experiments have allowed development of technical norms to guide forest operations according to site features (Table 1). First cleaning should be applied starting at 3–5 years for normal sites and, slightly later, 4–7 years (depending on site density) in the case of plantations on degraded sites.

The second intervention should occur after 3–5 years. Adequate results will be obtained when thinning operations are performed before stands reach an average diameter of about 20 cm (e.g., around 8–10 years old stands), and then this process is repeated every 4–6 years [21]. Intensity of tending operations (% of extracted trees), and especially that of thinning, plays an important role in the stand's further development with respect to growth and wood quality. Experimenting on the effects of type and intensity of thinning, Armasescu et al. [22] concluded that black locust reacts to thinning even at ages > 20; natural self-thinning continues after thinning operations; classical thinning (low and high)

does not lead to significant increase in total stand volume. The study also showed that very intensive thinning (>35% of total volume extracted) that lowers the stand density index below 0.8 actually leads to a decrease in stand production of up to 10%, in comparison to moderate thinning (25–30% of total volume extracted from all Kraft categories across the entire stand) which leads to an increase in stand productivity of up to 25%.

**Table 1.** Tending operations for black locust stands on degraded lands [20].

| Tending Operation Characteristics Depending on Stand Density | Normal Tending Operations (Trees $x$ ha$^{-1}$) | | Delayed Tending Operations (Trees $x$ ha$^{-1}$) | |
|---|---|---|---|---|
| | >5000 Trees $x$ ha$^{-1}$ | | | |
| Age of stand when tending is done (years) | 4–6 | 6–7 | 10–20 | 10–20 |
| Tending operation and intensity of extraction (% of volume or basal area) | Cleaning I | Cleaning I | Cleaning I | Cleaning I |
| | strong/very strong (16–30%) | strong (16–25%) | strong (16–25%) | moderate/strong (10–20%) |
| Trees per ha after the extraction Periodicity (years) | 4000–5000 3–5 | 3000–5000 3–5 | 2000–3500 3–5 | 2500–3000 3–5 |
| Tending operation and intensity of extraction (% of volume or basal area) | Cleaning II | Cleaning II | Cleaning II/Thinning I | Cleaning II/Thinning I |
| | strong (16–25%) | strong (16–25%) | strong (16–20%) | moderate (6–15%) |
| Trees per ha after the extraction Periodicity (years) | 2500–3000 5–7 | 2000–2500 5–7 | 2000–2500 5–7 | 2000–2500 5–7 |
| Tending operation and intensity of extraction (% of volume or basal area) | Cleaning III | Cleaning III | Cleaning III/Thinning II | Cleaning III/Thinning II |
| | moderate (6–15%) | moderate (6–15%) | moderate (6–15%) | moderate (6–15%) |
| Trees per ha after the extraction | 1500–2000 | 1500–2000 | 1500–2000 | 1500–2000 |

### 3.2.6. Stand Management

Forest management planning sets either productive or protective objectives for black locust forests, following strict management indicators: land use, silvicultural regime, stand structure, rotation, and harvesting technique. When dealing with such technical requirements, the forest management plan has to follow ecological, social, and economic local needs. While some stands' fate is wood production (mainly timber and construction wood), many stands have additional strong protective functions (e.g., degraded land reclamation, protection of soils and infrastructure, etc.).

### 3.2.7. Silvicultural Regime

In Romanian forestry law and official technical norms [19], the recommended form of management system for black locust stands is the coppice. Two forms can be differentiated: the simple one applied extensively and the selection coppice applied locally and occasionally. The simple coppice consists of clear-cutting of the stand followed by vegetative regeneration or seedling plantations. In the coppice selection system, a part of the sprouts/root-suckers from each stump can be preserved, eliminating only the crooked ones and those that have reached the targeted diameter. It can be applied with experimental purpose in the case of plantations on severely degraded lands or with small privately owned forests. In the past, some stands were managed as coppices with reserves, meaning that a percentage of the old stands were held until the following cycle in order to obtain trees with larger diameters [23].

### 3.2.8. Stand Composition and Structure

Black locust generally achieves large areas of continuous pure stands. Being a typical exclusivist species, stands are generally pure and even aged, their intraspecific competition being poor [24]. Nevertheless, on degraded lands, it is planted together with other exotic species.

Black locust can tolerate other species, most commonly honey locust (*Gleditsia triacanthos* L.) on sandy soils in the south and east of the country, or with black pine (*Pinus nigra* L.) on eroded slopes in the east of Romania [24,25]. It behaves as an exclusivist species for ground vegetation, black locust stands have very poor herbaceous diversity (e.g., frequent are *Urtica* sp., *Sambucus ebulus*).

### 3.2.9. Harvesting and Rotation/Cycle Length

Black locust stands have the shortest rotation length in Romanian forestry. Rotation length differs as a function of the forest primary purpose (wood production or protection). If the stand primarily has a production purpose (e.g., timber or construction wood), harvesting age varies between 15 years for coppices classified under the lowest production class to 35 years for the planted stands classified as the highest production class. If classified as having protection function (e.g., steep terrains prone to soil erosion), the harvesting age can reach 40 years in stands [19]. Wood harvesting occurs through clear-cuts in plots with a maximum area of 5 ha.

### *3.3. Genetics, Selection, and Tree Breeding*

An outstanding natural variety—*Robinia pseudoacacia* var. *oltenica*—was identified in the stands located in the south-west part of Romania (Oltenia region) by Bârlănescu, Costea, and Stoiculescu in 1966 [26]. Due to its valuable auxological and morphological characteristics [27,28], this variety was the subject of tree selection and breeding by grafting/cutting [29,30] and in vitro micropropagation. Later on, eco-physiological research by Bolea at al. [31] showed that this variety has both higher intensity of photosynthesis and tolerance to droughts, due to larger leaf area index (LAI), compared to common stands. Conservation and expansion of the Oltenica variety was continued in recent years by producing seedlings from cuttings, as maternal lineages of identified plus-trees [13].

Genetics research prior to 1990 focused on tree selection and breeding for enhancement of productivity and wood standing volume [29,32,33], as well as selection of valuable forest and beekeeping genotypes by hybridization. Trials consisting in testing different provenances in comparative cultures, identified "plus" trees and enhanced the number of clones and establishing vegetative propagation methods (e.g., grafting) for the valuable forms [34]. Currently, Romania has six qualified seed orchards with a combined area of 27 ha [35].

Recently, in vitro micropropagation—as the most modern technology used in tree genetics—achieved important results in selecting valuable forms of black locust via organogenesis and somatic embryogenesis [36,37]. Over the last two decades, black locust genetic research also focused on isolation, culture, and regeneration of protoplasts [37], and the influence of endogenic and exogenic factors on somatic embryogenesis [38,39] determined the genetic parameters of half-sib (free pollinated) and full-sib (controlled pollinated) black locust descendants [40]. Mirancea I. [36] tested black locust multiplication and rooting in vitro phases on different cultural media and hormonal balances, concluding that the optimum concentrations for which explants can multiply is 6 g/L for NaCl and 100 g/L for CaCO3. More recently, Băbeanu et al. [41] studied the isoperoxidase pattern of in vitro culture of *R. pseudoacacia* var. *oltenica* on various media, learning how isoperoxidase activity can be modulated through media. Further tests have showed the resistance of the plantlets obtained by micropropagation to severe ecological conditions of *R. pseudoacacia* var. *oltenica*, emphasizing how deuterium-depleted water-based media amplifies the caulogenesis.

### 3.4. Vulnerability of Black Locust

Systematic records of biotic and non-biotic factors and damage data in Romanian forests have been collected through the Forest National Survey and Forecast System operations since 1958. Based on such data, the main cryptogamic diseases affecting *Robinia pseudoacacia* L. were listed as follows [18]: virosis, *Fusarium* sp. and *Cuscuta* sp. attacks on seedlings, mildew (*Oidium* sp.), sooty mold (*Coniothyrium* sp., *Alternaria* sp., Cladosporium sp.), leaf spots (*Phleospora robiniae*) and tar spots on leaves (*Ectostroma* sp.), twig blight (*Pseudovalsa* sp.) and *Chorostate* sp. on young shoots, black spots (*Cucurbitaria* sp.) on old shoots, and wood-destroying fungi (*Hironela* sp, *Trametes* sp., *Phellinus* sp.).

A comprehensive inventory in the main black locust biotopes (seedling nurseries, young plantations, and mature stands), as well as complete description of biology and adequate means of control of all harmful insects of black locusts was achieved by [42]. Most infestation power was assessed for defoliating *Lymantria dispar* L. and fruit damaging *Etiella zinckenella* Tr. Later on, Trantescu et al. [43] concluded that it is not economically justified to spray insecticide except in the case of very valuable stands (e.g., seed orchards). *Semiothisa alternaria* Hb. (*Macaria alternaria* Hb.) was identified as an important pest for black locust [44] and it was estimated that the damage critical number by *Lymantria dispar* L. is 2–6 times higher for black locust stands than oak [45]. Information was brought forward on novel attacks in the south-western part of Romania of three leaf-miner moths of North American origin (*Parectopa robinella* and *Phyllonorycter robiniella* Clemens), accidentally introduced in Europe and apparently expanding their area [46,47].

Older stands show tree dieback in drought-prone areas, a fact apparently linked to higher cycles of vegetative regeneration and stands on poor sites [25,46]. Furthermore, dieback of black locust trees was reported in industrial zones, one example being near the town of Copșa Mică (in central Romania) due to soil and air pollution with sulfur and heavy metal compounds generated by industry (carbon black smoke) until the early 1990s. The main symptom was leaf necrosis, with the effects diminishing after the pollution source was removed in past decades [48–52].

Black locust bark leaves and young shoots are occasionally consumed by herbivores, such as game species, especially European brown hares and deer (roe-deer and red-deer). Domestic livestock (e.g., sheep, goats) consume leaves and young shoots on an occasional basis, with no national references regarding toxicity on animals or milk products. Locally, near villages—especially in private owned forests—black locust stands are under multiple human pressures such as illegal logging, erratic wood collection, and grazing [49].

Because plantations occur more often in low lands and drought prone areas, close to agricultural crops locations, black locust stands are subject to wildfires in which trees—even younger ones—survive while litter is burnt completely [51]. Long periods with very little/no rainfall were assumed as the main cause for dieback and decline of stands in 1980s and 1990s in the south and east of the country [53].

### 3.5. Wood Products and Other Uses of Black Locust
3.5.1. Wood Production

As a result of high percentage of coppice stands [54], the size of the black locust round wood usually does not exceed 20 cm in diameter at thin end, and quality of timber is relatively poor compared to key native species (Table 2). In most of the Romanian regions, short longevity because of the tree dieback is the main cause for rather small tree dimensions comparative to local broadleaved tree species. Therefore, majority of wood generated by black locust stands is mostly used as firewood or in rural construction (e.g., fences, sheds, poles) or less for props for gardening and vineyards [55]. The good quality wood can also be used for interior and garden furniture, parquet and floors or woodchip boards while more common uses are for fenceposts, poles, railroad crossties, stakes and fuel wood [56–58]. Traditionally, its wood is very appreciated for making barrels, handles for tools (e.g., axes, shovels, etc.).

**Table 2.** Black locust yield compared to main broadleaf species in Romania (for average yield production class III, pure, and even aged stands at 20 years old) [59].

| Species | Stand Structure | Age (Years) | Mean Diameter (cm) | Average Height (m) | Standing Volume (m³/ha) |
|---|---|---|---|---|---|
| *R. pseudoacacia* | coppice | 20 | 13.7 | 15.9 | 185 |
| *R. pseudoacacia* | plantation | 20 | 13.8 | 16.1 | 195 |
| *Quercus robur* | high stand | 20 | 8.0 | 8.3 | 97 |
| *Populus* sp. | High stand | 20 | 18.4 | 14.4 | 226 |
| *Fagus sylvatica* | high stand | 20 | 5.2 | 7 | 56 |

### 3.5.2. Non-Wood Products

Black locust is so appreciated by foresters and farmers alike due to a range of non-wood products that are available already at an early age of stands [60]. The most valued non-wood byproduct is honey, considered of the highest quality [61], which makes it the most expensive on the market. Compared to the traditional melliferous species (linden/lime, black locust, sunflower, rapeseed), black locust is the first to bloom (in May); therefore, beekeepers start the pastoral beekeeping in the black locust forests. Black locust stands have a high melliferous potential in Romania with up to 697,000 tonnes of honey per year. In order to ensure that the full benefit is reached, an application to support the planning of the pastoral beekeeping was developed based on forest maps, as a tool for decision makers at a national level in the planning of pastoral activity of the beekeepers. Black locust flowers show medicinal use in teas or infusions for digestive and pulmonary effects, and they have a calming effect on the nervous system [62–64]. A mix of flowers and leaves can be used as a tea drink to treat stomach pains and migraine. Generally, they have to be used in small quantities due to slight toxicity [65]. The seeds and bark of black locust are not used in traditional medicine because they contain substances that are toxic to humans [66].

### 3.6. Landscape Improvement Contribution

In Romania, black locust was very successful in reclaiming degraded lands [67] by exercising its anti-erosion role together with phytoremediation [68], and biomass accumulation in site conditions strongly prohibitive to other species [69]. On industrial or mining dumps, black locust plantations reached volumes of up to 73 m³/ha among 8-year-old stands [70]. It was also successfully used in the ecological remediation of historically heavily polluted lands [71]. In a Kyoto Protocol project of afforestation implemented in Romania, about 2500 ha were afforested with black locust in the south of the country to reclaim degraded and marginal agricultural lands [72]. Ten years after their afforestation, black locust plantations have reached heights of 14 m and basal diameters of 16 cm (Figure 2).

Biomass production and $CO_2$ sequestration of young trees was highlighted in recent years through allometric models of growth [73]. Black locust plantation biomass production on degraded lands in the south-west of Romania can reach 9.4 tons of dry mass per hectare at age 4, while coppiced stands average about 7.0 t dry mass ha$^{-1}$ [74].

Short rotation crops (SRC) for biomass energy use based on *Robinia pseudoacacia* represent a viable solution for degraded lands across Romania; however, a lack of state subsidies for afforestation investments by private land owners kept this branch relatively undeveloped [75]. Across Europe, recent studies have highlighted the increased use of black locust in SRC for bioenergy compared with the traditional species used such as poplar and willow. Among advantages of using Robinia are large biomass yields, high density wood, low moisture, and greater calorific output [76–79].

The potential of *Robinia pseudoacacia* plantations is significant in the efforts to mitigate the effects of climate change [74]. In the short term, the largest amount of C accumulation occurs in tree biomass, because the C is stored at least for the rotation period/life span of the forest, or even longer when manufactured into furniture, while litter and organic matter of mineral soils also act as steady growing carbon pools. Whether planted on degraded

agricultural lands [80,81] or as agroforestry shelterbelts [82], black locust C accumulation capacity is surpassed only by that of native and hybrid poplars (Table 3).

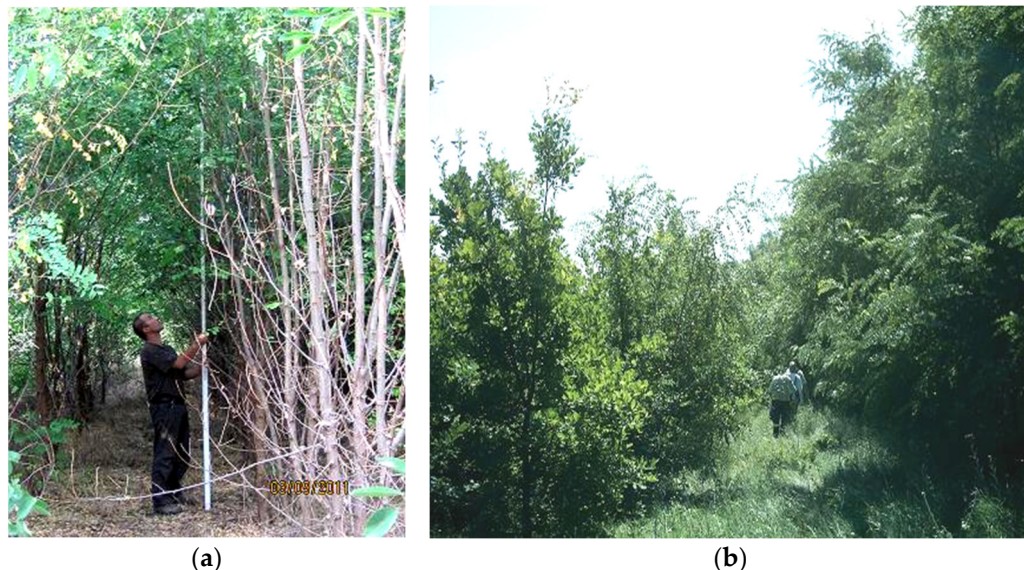

|        (**a**)        |        (**b**)        |

**Figure 2.** (**a**) Four-year-old black locust plantation; (**b**) Ten-year-old plantations of oak mix (left) and black locust (right).

**Table 3.** Carbon (tC ha$^{-1}$) stock in woody living biomass in plantations/shelterbelts for black locust and poplar.

| Age (Years) | C stock in Plantations on Degraded Agricultural Lands [82] | | C stock in Forest Shelterbelts [4] | |
|---|---|---|---|---|
| | *Robinia pseudoacacia* | *Populus* sp. | *Robinia pseudoacacia* | *Populus x euroamericana* |
| | (tC ha$^{-1}$) | (tC ha$^{-1}$) | (tC ha$^{-1}$) | (tC ha$^{-1}$) |
| 5 | 3.52 | 1.66 | 5.47 | 3.71 |
| 10 | 16.90 | 10.20 | 17.02 | 14.56 |
| 15 | 23.79 | 17.12 | - | 29.89 |
| 20 | 32,73 | 39,77 | 41 | 43.38 |
| 25 | - | - | 53.93 | 54.51 |
| 30 | - | - | 60.51 | 63.21 |

### 3.7. Ecosystem Services

When used on degraded lands, social benefits include improving local microclimate and mitigating the negative effects of climate change [83,84]. Shortly after planting, the stands start ensuring improvement of local biodiversity by offering shelter and food sources for birds, mammals, and other species. The economic and ecological role of black locust in agroforestry is represented by the forest shelterbelts to protect crop fields in the south of Romania [85–88], showing that the presence of shelterbelts in the Oltenia region led to an increase in different crop productions (e.g., wheat) of up to 130%, as opposed to unprotected fields. The shelterbelts are also a source of wood, honey, and they offer shelter for game and bird species. As one of the main species used in degraded land restoration, black locust plays an important role at both local and regional levels [13] in the effort to adapt and mitigate climate change by diversifying local supply for wood and feedstock or revenues from improved land use. Under recent more advanced ecosystem services payments, afforestation of degraded lands allowed further economic benefits, by trading greenhouse gas emissions reductions generated by tree plantations as financial instruments provided by the Kyoto Protocol [81]. Black locust is also appreciated in landscaping or

gardening, for its decorative grape-like—sweet perfumed—white flowers, and also for its robustness and adaptability [89].

*3.8. Invasiveness and Control of Black Locust*

In some European countries, the species is considered invasive [90]. Caused in general by inappropriate silvicultural decision-making and practices, the invasive character of black locust manifests itself as a result of the species great adaptability, highly developed vegetative regeneration (especially sprouting), soil condition alteration, and fast rate of growth compared to native species. When planted in close mixtures with slower growing species (e.g., oak) it overwhelmed them, after which it was extremely difficult and expensive to substitute it [91]. In this respect, Romanian forestry law and technical norms provide for the use of black locust only on degraded lands that prohibit the use of native species (severe site conditions). Its invasion on non-forest lands—e.g., orchards and vineyards especially—has occurred on lands abandoned after 1990. While, in other cases, it expands especially because of soil disturbance and stimulation of root-sucker growth. According to Enescu and Dănescu [92] "black locust should be regarded more as a very useful multi-purpose tree species with a high potential for forest land reclamation, rather than a dangerous invasive neophyte. Nevertheless, the presence of this species should be carefully monitored around nature reserves and fragile landscapes in nutrient-poor and dry locations."

The official status of black locust in the European Union is addressed in the EU Regulation 1143/2014 on invasive alien species [93] where black locust is included in the list of 80 invasive alien species (IAS) provided by the European Commission, but each EU country has to manage this species according to their national specificity.

## 4. Discussion

The knowledge regarding black locust has steadily progressed as the experience of Romanian foresters in cultivating this species increased. Nevertheless, research has brought forward the complexity and also vulnerability of this species that was considered to be the answer to more problems than it could solve.

The main strong point of black locust management in Romania is the successful use of the species to reclaim large areas of degraded lands [93–95], thus improving local landscape, biodiversity, and socioeconomic aspects (e.g., fuel and construction wood, beekeeping) for many rural communities [2,10,17,18].

Further improvement through selection/tree breeding is still needed in order to increase the quality of the plantation stands; however, unfortunately there is no ongoing program on this matter. Furthermore, forest managers have to take into account the importance and effects of periodic tending operations in black locust stands [28,29,35,37]. The demand on the market for material resulted from tending operations making the latter economically viable, thus it was implemented at the required development stages and intervals which in turn had a positive influence on stand growth and development. In these respects, applying cleaning and thinning operations at appropriate intervals could lead to increases in stand productivity of up to 25% [17,23].

The management mistakes of the past have revealed both the invasive ability of the species when planted extensively on productive sites in the detriment of native species, and also its limitations if planted in inappropriate site conditions (e.g., soils with excess humidity, high calcium carbonates) [10,92].

Being a relatively new species in the Romanian flora, black locust has few major threats; although this seems to be changing, especially due to more frequent migration of some allochthonous insects accidentally released in Europe [42–47].

The present work highlights the results of research carried out at national level on black locust management, vulnerability, and uses in the past 150 years in Romania. Although most papers cited in this review were published before 1990, this does not necessarily mean that this species presents less interest today for forestry researchers and practitioners.

Having in mind the large areas of degraded lands still existent in Romania, as well as the international efforts to support the use of renewable energy sources, black locust stands out as one of the most suitable species for the phytoremediation of these unproductive areas, thus helping to mitigate the negative effects of global climate change.

**Author Contributions:** Conceptualization, A.L.C.; Methodology, A.L.C.; Software, A.L. and C.M.; Validation, I.V.A. and L.D.; Formal analysis, A.L.C.; Investigation, A.L.C.; Data curation, A.L.C.; Writing—original draft preparation, A.L.C.; Writing—review and editing, C.G.C.; Visualization, C.G.C. and B.S.; Supervision, I.V.A. and L.D. All authors have read and agreed to the published version of the manuscript.

**Funding:** This research received no external funding.

**Institutional Review Board Statement:** Not applicable.

**Data Availability Statement:** Not applicable.

**Acknowledgments:** This paper was supported by the Sectoral Operational Programme Human Resources Development (SOP HRD), ID76945 financed from the European Social Fund and by the Romanian Government.

**Conflicts of Interest:** The authors declare no conflict of interest.

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
