# Peer review of "Black Locust (Robinia pseudoacacia L.) in Romanian Forestry"

_diversity, doi:10.3390/d14100780_

Round 1

Reviewer 1 Report

The manuscript comprehensively presents the relevant studies on black locust species in Romania, including the species distribution, species propagation, stand management and vulnerability issues. This review, in general, is highly summary and well structured, especially with the cultivation and management of black locust in Romania, and provides a detailed analysis of the management and application of black locust in the region. Although the authors mentioned the genetics, selection and tree breeding for black locust in Romania, genetic resources are the fundamental source for plant breeders selecting and improving desired traits through breeding, and perhaps the authors could add a review of relevant studies to give the reader a clearer understanding of the current status of propagation and breeding in Romania. In addition, although the authors mention modern techniques, including in vitro micropropagation, organogenesis and somatic embryogenesis, there is no mention of the prospects for the application of these modern techniques for the development of black locust in the discussion section.

 In line 63. “…… (mean annual temperatures above 100℃)”. It’s hard to understand. Perhaps the author wants to describe the annual accumulated temperature or effective accumulated temperature.

In line 137-140. “……while stand density influences the rate of growth, so that higher planting densities generate bigger growth rates.” This description is not very appropriate. Because of the afforestation density is too large, which can promote the height growth of trees and accelerate the growth speed of stand height. However, during the growth process, there will be certain competition among trees in terms of sunshine, water and nutrition. Therefore, the afforestation density will affect the growth of the diameter of trees. The density is large, the height of trees is thin, the pruning condition is good, the crown is narrow, and the physique of trees is poor.

The font format is not uniform including “Figure 1”, “Figure 2”, “Table 1”, “Table 2” and “Table 3”.

Author Response

Point 1. In the text was included new data and reference regarding the current status of propagation and breeding in Romania (line 213).

Point 2. Mean temperature value was corrected in line 63.

Point 3. Text was updated accordingly (lines 139-140). Higher stand densities are recommended only for stands located on degraded lands (e.g., sterile dumps) in order to fulfill their soil-ameliorative function while this is not the case for other stands due to the slenderness coefficient of the trees.

Point 4. The font size for tables and figures was corrected in the manuscript.

Reviewer 2 Report

The paper deals with Robinia pseudoacacia, a species of North-American origin, that in many European countries is considered invasive. Authors wrote a review paper about its role in Romanian forestry.

They were several monographic papers about black locust:

Vítková, M., Müllerová, J., Sádlo, J., Pergl, J., & Pyšek, P. (2017). Black locust (Robinia pseudoacacia) beloved and despised: A story of an invasive tree in Central Europe. Forest Ecology and Management384, 287-302.

Huntley, J. C. (1990). Robinia pseudoacacia L. black locust. Silvics of North America 2: 755-761.

Cierjacks, A., Kowarik, I., Joshi, J., Hempel, S., Ristow, M., von der Lippe, M., & Weber, E. (2013). Biological flora of the British Isles: Robinia pseudoacacia. Journal of Ecology 101(6): 1623-1640.

Rédei, K., Osvath-Bujtas, Z., & Veperdi, I. (2008). Black locust (Robinia pseudoacacia L.) improvement in Hungary: a review. Acta Silvatica et Lignaria Hungarica 4: 127-132.

Those papers and other ones that presented studies on R. pseudoacacia are almost absent (Cierjacks et al 2013 was cited). I understand the idea behind the paper and main aim, however, in my opinion papers from other countries except for Romania should be welcome.

In Material and methods authors should write how many Romanian scientific papers, reports, documents even personal communications from foresters were analysed. I suggest to put them in supplementary file.

The map with distribution of the species in Romania would be very helpful. Instead of that only distribution map of black locust stands in main culture regions of the country was presented.

In Results some information is mixed. For instance in chapter “Vulnerability of black locust”  there are mixture of biotic interactions of tree with fungi, reaction of the species to environmental pollution, reaction to herbivores etc.

I would prefer different division of the content i.e.: history of introduction, geographical area, habitat preferences, biotic interactions and then all other chapters that refer to management  followed by ecosystem services and invasiveness and control of black locust. In line 355 there is “Invazivity” and should be replaced by “invasiveness”.

 Discussion must be extended and could focus on differences in species behaviour between Romania and native range as well as other countries where black locust is introduced species. The other important aspect that partially was discussed is the invasive status of the species. How harmful or/and beneficial R. pseudoacacia is in Romania?

Author Response

Point 1. As mentioned by the reviewer, there are other monographic papers about black locust and most of them are referring to this species from a national point of view.

Point 2. Aditional data regarding the source and number of publications was added in the Materials and Methods chapter (lines 48-50)

Point 3. The distribution map for black locust in main culture regions was the result of the efforts of the authors team. There is a map of the Romanian forests by ecosystem types but its resolution prohibits the identification of black locust stands in an article format.

Point 4. History of introduction, geographical area, habitat preferences were the subject of a previously published article (https://www.cabi.org/ISC/FullTextPDF/2014/20143263765.pdf). The text was updated (term was corrected) in line 355.

Point 5. Due to the large size of the paper only the essential aspects were touched in the Discussion chapter. From the content of the whole manuscript and highlighted in the Discussion chapter is the fact that although black locust has the potential to harm natural habitats, due to the fact that is was almost exclusively used on degraded lands in Romania it was very beneficial for the phytoremediation of large areas.

Reviewer 3 Report

This manuscript is well written, but it is not the first review on this tree species, unfortunately! Two years ago Nicolescu et al. published a much broader review (Ecology, growth and management of black locust (Robinia pseudoacacia L.), a non‑native species integrated into European forests). Although your manuscript focuses on Romanian forests a lot of information you give on black locust is rather common. So in general, I have some doubts that your paper is innovative enough. Before reviewing in detail a decision must be made by the editors whether this reviewing process should be continued or not!

Author Response

Point 1. Not that it matters, but I wrote this manuscript 10 years ago, yet never got to the point of ''cleaning it up'' and publishing it. As for the innovative aspect...its a review article.

Round 2

Reviewer 3 Report

Dear authors,

I appreciate your improvements in the manuscript, but writing a manuscript years ago does not mean that adding new literature is impossible! So I still believe that this could in general be improved.

Some minor things:

L. 269: It must be "European brown hares"

L. 624-626: Wrong way of presenting, adapt to other format!

Author Response

Point 1: ''I appreciate your improvements in the manuscript, but writing a manuscript years ago does not mean that adding new literature is impossible! So I still believe that this could in general be improved.''

 Response 1: We incorporated in the manuscript text (e.g., lines 106, 107) data from 3 new references (ref. no. 94, 96, 97) on research related to black locust stands in Romania, published internationally between 2017 to 2020.

Point 2: ‘’Some minor things:

  1. 269: It must be "European brown hares"
  2. 624-626: Wrong way of presenting, adapt to other format!’’

Response 2:  The manuscript text was corrected accordingly.
